# Unsupervised Domain Adaptation for Medical Image Segmentation via Self-Training of Early Features

**Rasha Sheikh**                                      RASHA@CS.UNI-BONN.DE
**Thomas Schultz**                                    SCHULTZ@CS.UNI-BONN.DE
*University of Bonn*

## Abstract

U-Net models provide a state-of-the-art approach for medical image segmentation, but their accuracy is often reduced when training and test images come from different domains, such as different scanners. Recent work suggests that, when limited supervision is available for domain adaptation, early U-Net layers benefit the most from a refinement. This motivates our proposed approach for self-supervised refinement, which does not require any manual annotations, but instead refines early layers based on the richer, higher-level information that is derived in later layers of the U-Net. This is achieved by adding a segmentation head for early features, and using the final predictions of the network as pseudo-labels for refinement. This strategy reduces detrimental effects of imperfection in the pseudo-labels, which are unavoidable given the domain shift, by retaining their probabilistic nature and restricting the refinement to early layers. Experiments on two medical image segmentation tasks confirm the effectiveness of this approach, even in a one-shot setting, and compare favorably to a baseline method for unsupervised domain adaptation.

**Keywords:** Unsupervised Domain Adaptation, Segmentation

## 1. Introduction

Annotating medical images to supervise the training of deep neural networks for segmentation is time-consuming and often requires medical experts. Once trained, these models perform well on similar data from the same site, but the performance often drops on data acquired in a different site with another type of scanner for example. Transfer learning and domain adaptation offer various solutions to this problem by adapting the model to new data and addressing the domain shift between source and target domains. Unsupervised domain adaptation attempts to do so without using any labeled target data.

There are different approaches to unsupervised domain adaptation for the task of semantic segmentation. Several works use an adversarial scheme to learn domain-invariant features (Hoffman et al., 2016) or image-to-image translation as in CycleGAN (Zhu et al., 2017) to adapt the segmentation model (Li et al., 2019). Others perform a layer-wise matching of activations between the domains (Huang et al., 2018), or aim to minimize the entropy of target predictions (Vu et al., 2019) based on the observation that source predictions often have higher confidence values.

Our proposed approach follows a self-supervision strategy. Self-supervision either makes use of auxiliary tasks (Sun et al., 2019) or losses (Hu et al., 2021) that do not require supervision, or it refines the network based on its own predictions on the target domain. An important issue with the latter strategy is to avoid propagating incorrect predictions.

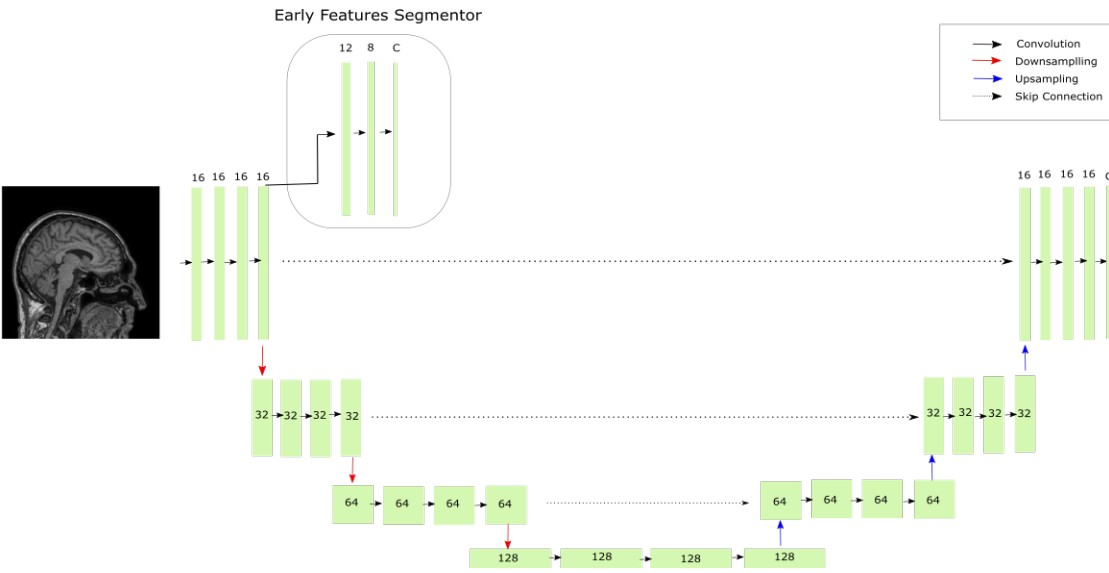

Figure 1: Architecture of the segmentation model.

This has been approached by filtering predictions so that only the most confident ones are used as a training signal (Zou et al., 2018). We propose to use the predictions differently, inspired by recent work that has observed that the first layers of U-Net models learn more domain-specific features (Shirokikh et al., 2020), and benefit most from a refinement when limited training data is available (Zakazov et al., 2021).

Therefore, we add a segmentation head after the first few convolutional layers and use final predictions of the network as pseudo-labels to refine only those early features. Our main finding is that this leads to a stronger improvement of the final segmentation than a filtered self-training of the whole network. We believe that this reflects a greatly decreased re-enforcement of incorrect predictions, because our pseudo-labels retain their probabilistic nature, and because we limit adaptation to early layers. Our code is publicly available at https://github.com/ferasha/UDAS.

## 2. Method

### 2.1. Base Model

We first train a segmentation model on a source domain, and later adapt it to a target domain with unlabeled data. The base model is shown in Figure 1. It has a U-Net (Ronneberger et al., 2015) structure identical to the one used by (Shirokikh et al., 2020) where images go through a number of convolutional blocks that learn $M$ feature maps with the same spatial size as the input before proceeding with the encoder-like part of the architecture. It uses $3 \times 3$ convolution kernels, ReLU activation functions, and the skip connections are implemented as convolutions followed by a sum operation. The model is trained with the cross-entropy loss,

$$l(x, y)_{source} = -\frac{1}{N_S} \sum_{n=1}^{N_S} \sum_{c=1}^{C} y_{n,c} \log p_c(x_n), \tag{1}$$

where $N_S$ is the number of samples in the source images, $p_c(x_n)$ is our model's predicted probability that pixel $n$ in image $x$ belongs to class $c$, and $y_n$ is a one-hot-encoding vector of the true label for pixel $n$. We use the Adam optimizer, no augmentation, and 0.001 as the learning rate.

## 2.2. Domain Adaptation Through Self-Training

To prepare adapting the base model to the target domain, we first add another segmentation head to it, just before the first downsampling operation. This block is titled *Early Features Segmentor* in Figure 1. During refinement, this head will act as a student which is trained by the output of the overall base model, which acts as a teacher. Because student and teacher share the first few convolutional layers, refining the student on the target domain also benefits the teacher. While initializing the student on the source domain, we freeze all weights in the base model, and train with the same loss as in Equation 1, again using the ground-truth segmentation masks.

The model now produces two probabilistic segmentation outputs, a weak one $\tilde{p}_c$ based on early features, and the final segmentation $p_c$ at the end of the network. For domain adaptation, we feed samples from the target domain through both branches of the network, and compute a cross-entropy loss between them,

$$l(x)_{target} = -\frac{1}{N_T} \sum_{n=1}^{N_T} \sum_{c=1}^{C} p_c(x_n) \log \tilde{p}_c(x_n), \tag{2}$$

where $N_T$ is the number of samples in the target images.

We now use the stronger segmentation $p_c$ at the end of the network to improve the weaker early segmentation $\tilde{p}_c$. This adapts the early features based on the richer and higher level information that was learned by the rest of the network. In this phase, we only update the weights of the early convolutional blocks that are shared between the two branches, freezing the weights in the early segmentation head itself. Since we minimize Equation 2 only with respect to the early segmentation $\tilde{p}_c$, we do not obtain a gradient in the rest of the U-Net, so weights there remain unaffected as well. Despite this, the probabilities $p_c$ that we use as pseudo ground truth also change during the refinement process, since they are affected by the updates in the early layers. Throughout the refinement, we track the Dice score between the early and final segmentations. We stop when the absolute difference between the Dice in the current and the previous epoch drops below 0.005.

## 3. Experiments

### 3.1. Calgary-Campinas Dataset

The Calgary-Campinas dataset (Souza et al., 2018) consists of 359 3D volumes of brain MR images with corresponding skull-stripping segmentation masks. The data is generated

Table 1: Surface Dice scores on the Calgary-Campinas dataset. ST and CBST refer to the self-training and class-balanced self-training proposed by (Zou et al., 2018).

|             | Base Model | ST      | CBST    | Ours       |
|-------------|------------|---------|---------|------------|
| GE 1.5      | 0.55487    | 0.53042 | 0.55347 | **0.75887** |
| Philips 1.5 | 0.74974    | 0.72526 | 0.77563 | **0.84601** |
| Philips 3   | 0.65806    | 0.66237 | 0.68958 | **0.85810** |
| Siemens 1.5 | 0.70478    | 0.69294 | 0.75003 | **0.82457** |
| Siemens 3   | 0.88651    | **0.89180** | 0.88418 | 0.88740 |

using six scanners which differ in the vendor type and the field strength. Those scanners represent the different domains in our experiments.

We train on 40 subjects from GE 3 (i.e. source domain) and test on 10 subjects from each of the other target domains. The only pre-processing is a min-max scaling of each volume. To evaluate the performance, we follow (Shirokikh et al., 2020) and use the surface Dice score (Nikolov et al., 2018), which quantifies the fraction of the predicted and ground truth surfaces that are within a pre-specified distance of each other. This score is deemed more informative than the usual volumetric Dice in this context because it focuses on the structure of interest, i.e., the brain contour, as opposed to the large, but mostly trivial internal volume.

### 3.1.1. IMPROVEMENT OVER BASE MODEL

Table 1 shows the adaptation result on the different domains. Here, the label "base model" refers to the model that was trained on the source domain, without any adaptation. Our approach significantly improves upon the base model. Qualitative results from all targets domains are shown in Figure 2.

To further validate our approach, we also treated another domain, Siemens 3, as our source domain and adapt to the other target domains. These results are shown in Appendix A. Once again, we observe an improvement in the performance.

To illustrate that the refinement works as described above, Appendix B shows the early and final probabilistic segmentations for an example input, before and after refinement.

### 3.1.2. COMPARISON TO PREVIOUS WORK

We consider the previous work by (Zou et al., 2018) to be most similar to ours, since it also uses a segmentation loss towards pseudo-labels to refine the network weights. Unlike our approach, they filter the predictions of the network to keep those with higher confidence values and use them as labels to refine the whole network. They implement two variations of this idea, with and without class-balanced filtering. These are referred to as Class-Balanced Self-Training (CBST) and Self-Training (ST) in Table 1. On our data, we observe moderate improvements with this approach and occasionally a drop in performance. For a more direct comparison, we also tried to restrict the CBST approach to update the same layers as our method. However, results were worse than when refining the entire model.

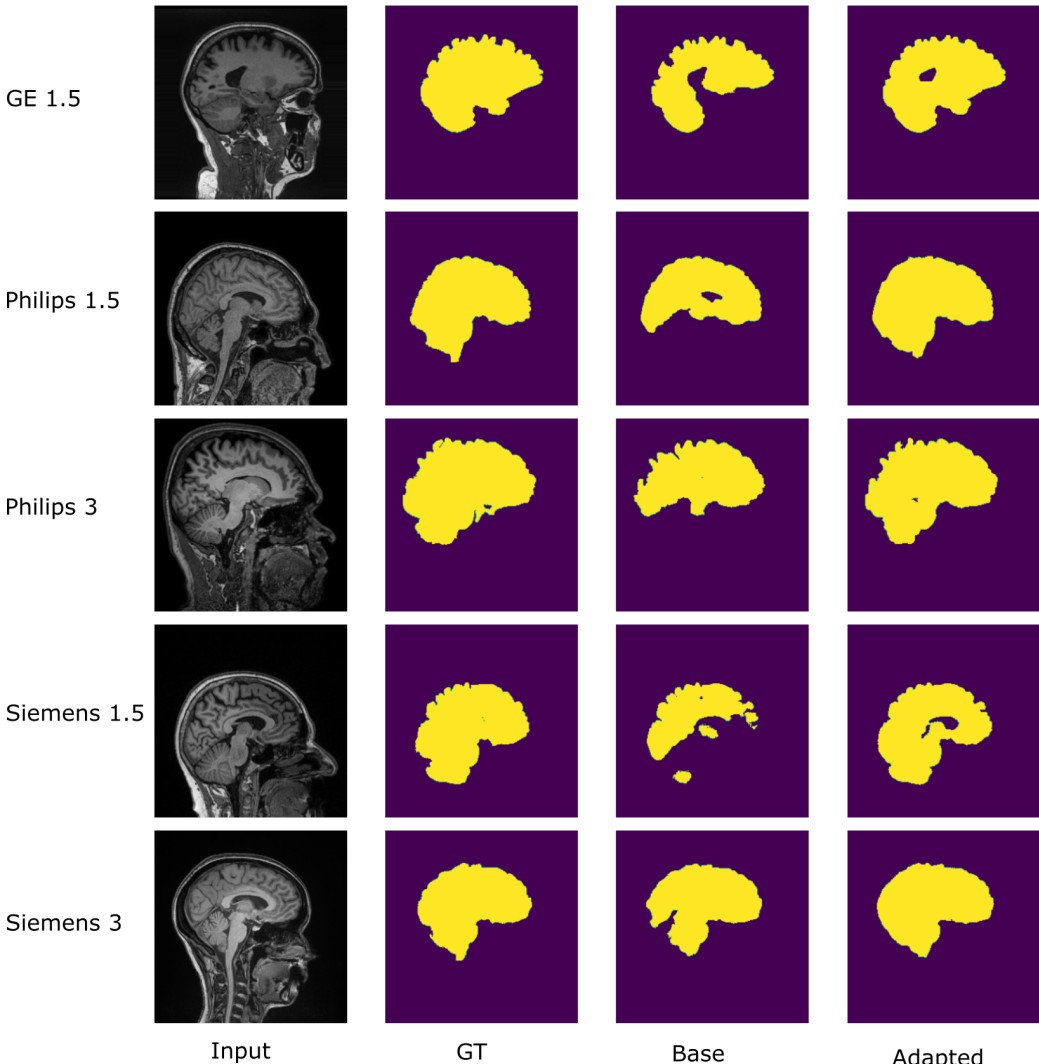

Figure 2: Qualitative results from the Calgary-Campinas dataset. Columns show the input image, ground-truth, and segmentation using the base and adapted models, respectively. The rows represent the different domains.

Table 2: Surface Dice when refining and testing on one target subject at a time, averaged over 10 subjects.

|  | Base Model | One-Shot Refinement |
| --- | --- | --- |
| GE 1.5 | 0.55487 | **0.73655** |
| Philips 1.5 | 0.74974 | **0.84003** |
| Philips 3 | 0.65806 | **0.85312** |
| Siemens 1.5 | 0.70478 | **0.82449** |
| Siemens 3 | **0.88651** | 0.87900 |

Table 3: Surface Dice of the adapted model on a new subset from the same target domain, without additional refinement.

|  | Base Model | Fixed Adapted Model |
| --- | --- | --- |
| GE 1.5 | 0.46892 | **0.72773** |
| Philips 1.5 | 0.78818 | **0.88231** |
| Philips 3 | 0.55264 | **0.82302** |
| Siemens 1.5 | 0.72790 | **0.82344** |
| Siemens 3 | 0.88564 | **0.90815** |

### 3.1.3. One-Shot Domain Adaptation

In the one-shot setting, a single dataset should be segmented, and is the only data available from the target domain. Running our refinement in that mode, separately for each subject in our test set, produced results that were almost as accurate as the refinement on 10 volumes, which was reported above. Table 2 shows the resulting average surface Dice score.

### 3.1.4. Generalization to Unseen Data

After using our approach to refine the early layers on 10 subjects from the target domain, we applied the model to further data from the same domain without further refinement. Results on a subset of the target domain that was disjoint from the one used for refinement are shown in Table 3. They confirm that the model now generalizes successfully.

### 3.1.5. Alternative Modes of Refinement

Given the benefit from refining the first few convolutional layers, it is natural to try a similar strategy for refining deeper layers as well. Due to the mismatch in image resolution at deeper layers, this requires a resampling of features, predictions, or labels. Even the variant that worked best in our experiments only provided a marginal additional benefit when compared to the simpler refinement of the earliest layers alone.

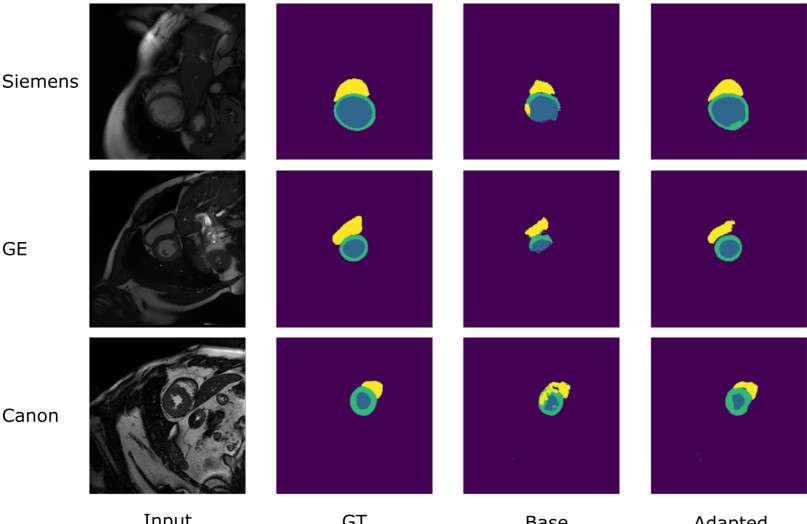

Figure 3: Qualitative results from the M&Ms dataset. The rows represent the different domains. The columns show the input image, ground-truth, segmentation using the base and adapted model respectively. Yellow: RV, Blue: LV, Green: MYO

Moreover, we tried refining only the batch normalization layers, as proposed for domain adaptation by (Hu et al., 2021). However, refining all weights gave better results in our case. Experimental results related to these alternatives are discussed in Appendix C.

### 3.2. Multi-Centre, Multi-Vendor & Multi-Disease Cardiac Image Segmentation (M&Ms) Dataset

We also present results on a more challenging multi-class segmentation task. This dataset (Campello et al., 2021) consists of four domains corresponding to images from four scanner vendors. The data from the different sites vary in their in-plane resolution, slice thickness, number of slices, and number of time frames.

The publicly available data includes cardiac MR scans of 345 subjects. The segmented regions are the left ventricle cavity (LV), the right ventricle cavity (RV), and the left ventricle myocardium (MYO). The only pre-processing we apply is min-max scaling.

We train on 75 subjects from the Philips training set (source domain) and test on the official test sets of the other target domains. We chose Philips as our source domain because of the large drop in performance when testing on the other domains (Campello et al., 2021). Exemplary segmentation results are shown in Figure 3. They illustrate differences in image appearance across the domains (rows), and the benefit of our refinement (final column) compared to the base model with respect to the ground truth (GT).

To quantify the performance, we use the volumetric Dice score

$$\text{Dice} = \frac{2 \sum \hat{y} y}{\sum \hat{y} + \sum y}, \tag{3}$$

Table 4: Volumetric Dice scores on the M&Ms Dataset. ST and CBST again refer to the self-training and class-balanced self-training proposed by (Zou et al., 2018).

|  | Base Model | ST | CBST | Ours |
|---|---|---|---|---|
| Siemens | 0.61926 | 0.54632 | 0.53262 | **0.68608** |
| GE | 0.43403 | 0.37390 | 0.38717 | **0.67550** |
| Canon | 0.65464 | 0.66477 | 0.68268 | **0.70576** |

where $\hat{y}$ and $y$ are the predicted and true labels respectively. Table 4 reports the corresponding results on the different domains. We again compare the performance to the self-training approach in (Zou et al., 2018) and also show the baseline performance with no adaptation.

We found that on this more challenging dataset, the selection made by ST and CBST often includes samples that are incorrect despite a high confidence. In our experiments, the accuracy among pixels that were selected for adaptation was sometimes as low as 20%. This explains why the filtering approach was sometimes detrimental on this dataset. In contrast to this, our use of probabilistic pseudo-labels from the full image for early feature refinement still provided a benefit.

Appendix D provides a breakdown of the per-class performance, demonstrating that all classes benefit from the refinement.

## 4. Conclusion

Domain shift frequently occurs in medical imaging when data generation differs between sites, e.g., when different scanners are in use. This can severely impact the performance of segmentation models on test data from a different site. Therefore, models have to be adapted. Unsupervised domain adaptation is an attractive solution since it lifts the need for annotating data from the other domain.

We proposed a novel, simple, and efficient strategy for domain adaptation via self-training and demonstrated clear qualitative and quantitative benefits on segmentation performance on two medical image segmentation tasks. We achieved superior performance compared to the CT and CBST baselines, which can be explained by reducing detrimental effects of propagating incorrect labels by retaining probabilistic pseudo-labels, and restricting the refinement to early layers. Using pseudo-labels and having to refine only a subset of weights also leads to fast training times: Our experiments only required up to five epochs. Compared to unsupervised domain adaptation based on adversarial training, our approach is easier to use because it does not require a careful balancing of the training signals from a generator and discriminator.

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

## Appendix A. Results With Siemens 3 as the Source Domain

Table 5: Surface Dice when using the Siemens 3T domain of the Calgary-Campinas dataset as the source domain.

|  | Base Model | Adapted Model |
|---|---|---|
| GE 1.5 | 0.60650 | **0.74609** |
| GE 3 | 0.82829 | **0.95561** |
| Philips 1.5 | 0.73747 | **0.83830** |
| Philips 3 | 0.56104 | **0.85981** |
| Siemens 1.5 | 0.47547 | **0.82672** |

## Appendix B. Illustration of Early and Final Segmentations

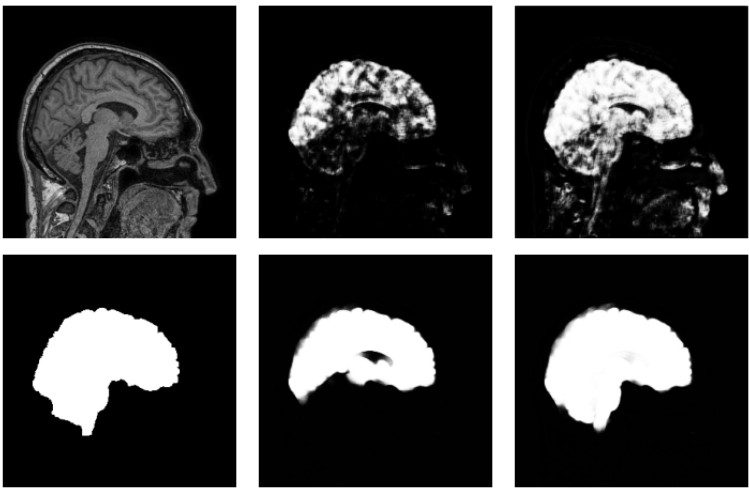

Figure 4: First column shows the input and ground truth. Second column shows the early and final segmentations using the base model. Third column shows the early and final segmentations using the refined model.

Figure 4 compares segmentations from the base model in the second column to the refined model in the third column. As expected given the limited receptive field and complexity of early features, the early segmentations (top row) are much weaker than the ones at the end of the network (bottom row). The top image in the third column shows that early feature refinement improved the early segmentation. The bottom image shows the final improvement when propagating the refined features through the remaining network.

## Appendix C. Results from Alternative Refinement Strategies

Table 6: Surface Dice when also refining deeper layers, or only batch normalization weights

|  | Base Model | Refine L1 | Refine L1 then L2 | Refine L1 and L2 | Refine L1-BN |
|---|---|---|---|---|---|
| GE 1.5 | 0.55487 | 0.75887 | 0.75994 | **0.76352** | 0.70093 |
| Philips 1.5 | 0.74974 | 0.84601 | 0.84120 | **0.85202** | 0.83101 |
| Philips 3 | 0.65806 | 0.85810 | 0.85979 | **0.86574** | 0.85083 |
| Siemens 1.5 | 0.70478 | **0.82457** | 0.82176 | 0.82338 | 0.81101 |
| Siemens 3 | 0.88651 | **0.88740** | 0.87205 | 0.87971 | 0.88291 |

Table 6 compares results from our proposed refinement ("refine L1") to alternative refinement strategies. In "refine L1 then L2", we extended our proposed method with a second refinement, in which we place an additional early segmentation head at the second resolution level, just before the second downsampling. We initialized it in the same way as it is described in Section 2.2, and used it to refine the weights at the second resolution level of the encoder, freezing the weights at the first level, which had already been refined previously. We resolved the resolution mismatch by upsampling logits and computing the segmentation losses at the original resolution. In "refine L1 and L2", we only used an early segmentation head at the second level, to update the weights of the first and second layer jointly. Compared to the benefit from refining the initial layers, the additional benefit from refining deeper layers with our method was marginal.

"Refine L1-BN" corresponds to our proposed method, but only refines parameters in the batch normalization blocks of the first layer. It did not perform as well as a full refinement of all weights.

## Appendix D. Class-Wise Quantitative Results on M&Ms Dataset

Table 7: Breakdown of class-specific Dice scores on the M&Ms Dataset

|  |  | LV | MYO | RV |
|---|---|---|---|---|
| Siemens | Base Model | 0.71047 | 0.54531 | 0.60199 |
|  | Ours | **0.77685** | **0.62302** | **0.65838** |
| GE | Base Model | 0.49160 | 0.38219 | 0.42831 |
|  | Ours | **0.74438** | **0.62778** | **0.65434** |
| Canon | Base Model | 0.71849 | 0.63117 | 0.61427 |
|  | Ours | **0.77674** | **0.65661** | **0.68394** |

