# OpenReview forum: "Unsupervised Domain Adaptation for Medical Image Segmentation via Self-Training of Early Features"
_MIDL.io/2022/Conference — MIDL 2022_

### Official Review · Reviewer_Kdin · 2022-01-22

**Confidence:** 4
**Preliminary Rating:** 3
**Recommendation:** Oral, Poster

**Summary:**

This paper presents a self-training method for unsupervised domain adaptation. With a modified dual-branch network, the network is able to conduct early layer fine-tuning using target images only. The method has been validated on two MR image datasets, with improved performance compared to a baseline method without adaptation. The results look very impressive. However, I have concerns with the design of the methodology and the fairness of the comparison study.

**Strengths:**

1. This structure of the paper is good.
2. The idea of modifying a U-net into a dual branch network architecture for self-training is somewhat interesting.
3. This method has been validated on two medical datasets (brain and cardiac). The performance gain on target domains is impressive.

**Weaknesses:**

1. Design choice: Why only adapt the features in the first conv block?  Have the authors tried other subsequent blocks? Recent related work has suggested that layers in the encoder part (not necessarily to be the first layers) are more likely to be tuned than those in the decoder for optimal finetuning [1].
2. This work is very similar to related works on test time adaptation, which are mainly focusing on fine-tuning batch normalization layers, e.g. [2].  It also has strong connections to recent works on self-supervised learning for unsupervised domain adaptation, which also adopts a dual-branch structure [3]. A comparison is needed to highlight the contribution.
3. It is not very clear to me about the self-training procedure. Does it look like both Eq 1 and Eq 2 are losses computed over N images for training?  Does it mean that this method requires equal sizes of source images and target images for source domain training and unsupervised target domain fine-tuning? I am also wondering if the proposed method still works when there is only one image in the target domain.
4. Comparison study:
    - For baseline methods ST and CBST, do the authors fix the latter layers before self-training for fair comparison? I feel the confidence thresholding in baseline methods looks more reasonable to me. Have the authors adopted the confidence-thresholding loss functions in their own proposed architecture for fair comparison?

References:
1. [https://link.springer.com/chapter/10.1007%2F978-3-030-87199-4_20](https://link.springer.com/chapter/10.1007%2F978-3-030-87199-4_20)

2. [https://link.springer.com/chapter/10.1007%2F978-3-030-87199-4_24](https://link.springer.com/chapter/10.1007%2F978-3-030-87199-4_24)

3. [https://arxiv.org/abs/1909.11825](https://arxiv.org/abs/1909.11825)

**Deanonymize Review:**

no

**Final Rating After The Rebuttal:**

4: Weak Accept

**Justification Of The Final Rating:**

The authors have successfully cleared all my concerns. The additional experimental results verify the effectiveness of the proposed method. Yet, the paper can be improved by adding a direct comparison to more related works. Given the simplicity and effectiveness of the proposed method, I believe this deserves acceptance.

**Paper Type:**

methodological development

**Questions To Address In The Rebuttal:**

1. For baseline methods ST and CBST, do the authors fix the latter layers before self-training for fair comparison? I feel the confidence thresholding in baseline methods looks more reasonable to me. Have the authors adopted the confidence-thresholding loss functions in their own proposed architecture for fair comparison?
2. Why only adapt the features in the first conv block?  Have the authors tried other subsequent blocks? Recent related work has suggested that layers in the encoder part (not necessarily to be the first layers) are more likely to be tuned than those in the decoder for optimal finetuning [1].
3. This work is very similar to related works on test time adaptation, which are mainly focusing on fine-tuning batch normalization layers, e.g. [2].  It also has strong connections to recent works on self-supervised learning for unsupervised domain adaptation, which also adopts a dual-branch structure [3]. A comparison is needed to highlight the contribution.


References:
1. [https://link.springer.com/chapter/10.1007%2F978-3-030-87199-4_20](https://link.springer.com/chapter/10.1007%2F978-3-030-87199-4_20)

2. [https://link.springer.com/chapter/10.1007%2F978-3-030-87199-4_24](https://link.springer.com/chapter/10.1007%2F978-3-030-87199-4_24)

3. [https://arxiv.org/abs/1909.11825](https://arxiv.org/abs/1909.11825)

**Special Issue:**

no

---

### Official Review · Reviewer_pCQT · 2022-01-24

**Confidence:** 4
**Preliminary Rating:** 5
**Recommendation:** Best Paper Award

**Summary:**

This paper concerns the adaptation of a UNet [Ronnenberger et al. 2015] by self-training on a new domain. It adds an early segmentation part, and a mixed training system. The method firstly trains a UNet, then it freezes the weights and adds a small 'head' which takes the output of the first UNet block and trains this network's head parameter only for early predictions. For a new domain, the output of the full is then used to train the early prediction model but only their shared first block of the UNet is updated. The method is tested on the Calgary-Campinas dataset [Souza et al., 2018] and on a Multi-class dataset [Campello et al., 2021], and the adaptation strategy gives considerable improvements over no adaptation.

**Strengths:**

The paper is well organized and written, the model is an appreciatively simple addition to the UNet, and although the training scheme is a bit complicated, the results are impressively improved without added knowledge of the new data.

**Weaknesses:**

I have little to complain about. The paper could make the training scheme more clear - I had to read it twice to understand it, and I don't think my recap above was very successful. I also think it would be interesting to understand the relationship between the expression power of the very limited 'head' model versus the full model, as well as see the change of the filters in the first block of the UNet. Also, the relation to p and \tilde{p} in (1) and (2) to the model in Figure 1 could be improved.

**Deanonymize Review:**

yes

**Detailed Comments:**

I have nothing to add here.

**Paper Type:**

methodological development

**Questions To Address In The Rebuttal:**

Sorry, I must write 200 characters: I don't think the authors need do more. I don't think the authors need do more. I don't think the authors need do more. I don't think the authors need do more. I don't think the authors need do more.

**Special Issue:**

yes

---

### Official Review · Reviewer_Xx7h · 2022-01-28

**Confidence:** 4
**Preliminary Rating:** 3
**Recommendation:** Poster

**Summary:**

The author applied self-supervised to refinement the early feature of U-net for domain adaptation in medical image segmentation. The approach is simple yet efficient. Experiments show that the proposed method outperforms other methods on two medical segmentation tasks, namely single class and multi classes segmentation.

**Strengths:**

- The method is simple yet efficient to solve the problem of domain adaptation.
- The experiment solves that it outperforms other methods on two different tasks, namely single class and multi classes segmentation.
- Clear organize of experiments


**Weaknesses:**

- There are minor grammar errors. Thus, the manuscript is hard to follow. The manuscript should be proofread and polished.
- I think that where to put the segmentation head is needed to be analysed. Is putting before the first downsampling operation the best solution?
- Training procedure is not clearly explained. Therefore, it is not easy to follow.


**Deanonymize Review:**

yes

**Detailed Comments:**

There are a lot of minor grammar errors, for example:
- Page 3, line 4 should not be indented; page 7, line 6 should not be indented.
- There should be “.” after equation (2)


**Final Rating After The Rebuttal:**

4: Weak Accept

**Justification Of The Final Rating:**

The authors gave feedback on all of my concerns. Although the location of the segmentation head is not thoroughly studied, I think that the manuscript deserves acceptance. Therefore, I would upgrade my final rating

**Paper Type:**

methodological development

**Questions To Address In The Rebuttal:**

- It is not clear if the weight of the first layer of U-Net is being updated when predicting the pseudo label. And why?
- Is putting segmentation head before the first downsampling operation the best solution? The authors may need to provide the experiment result to provide the conclusion


**Special Issue:**

no

---

### Meta-Review · Area_Chair_v8JK · 2022-02-20

**Recommendation:** Accept (Poster)
**Confidence:** 5

**Metareview:**

This paper proposes a domain adaptation technique for medical image segmentation using self-supervised refinement of early features of U-net.  With a modified dual-branch network, the network is able to conduct early layer fine-tuning using target images only. All reviewers agreed that the method is simple but effective, and the  results are improved without added knowledge of the new data. The questions from the reviewers about clarity of presentation, location of segmentation head and comparison to other methods have been sufficiently addressed during the rebuttal period and all reviewers agreed that this paper should be accepted. I agree with the reviewers and suggest acceptance of this paper.

.

---

### Decision · Program_Chairs · 2022-02-28

Accept